# Flatness-Based Backstepping Antisway Control of Underactuated Crane Systems under Wind Disturbance

Zian Yu and Wangqiang Niu *

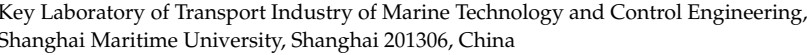

Key Laboratory of Transport Industry of Marine Technology and Control Engineering,
Shanghai Maritime University, Shanghai 201306, China
* Correspondence: wqniu@shmtu.edu.cn

**Abstract:** A control method that combines trajectory planning and backstepping is proposed for the antisway problem of underactuated overhead cranes under wind disturbance. First, a set of flat outputs is proposed so that the crane system dynamics can be represented by each order of flat outputs. Sufficient relevant constraints are given to ensure that the trolley can arrive at the desired position in a limited time under variable rope lengths, and that the swing angle can be suppressed when the payload is lifted or lowered during operation. The planned trajectory is obtained by solving for the optimal parameters of the flat output. Next, to reduce the deviation caused by wind disturbance on the actual control of the trajectory, a tracking controller is designed. Because the system output space and flat output space are differentiable homeomorphisms, the backstepping controller constructed based on the flat output can indirectly control the system output, which makes the backstepping method applicable to underactuated cranes. The simulation results show that the proposed method is effective and has strong robustness.

**Keywords:** differential flatness; underactuated systems; backstepping; trajectory planning; antisway

## 1. Introduction

Overhead cranes are widely used for cargo transportation in factories, warehouses, ports, and other places; thus, they have higher requirements for their transport efficiency. Because the crane payload swing has a direct impact on its transport efficiency and the safety of the transport process, the study of the crane's antiswing control is of practical significance.

The underactuated characteristics of the overhead crane make it more difficult to control. At present, some open-loop methods, such as input shaping [1–3] and trajectory planning [4,5], and closed-loop methods, such as sliding mode control [6–8], linear quadratic regulator (LQR) control [9–11], model predictive control [12–14], and proportional-integral-derivative (PID) control [15–17], have been proposed to solve the crane's antisway control problem. The design difficulty mainly lies in how to achieve the control of the swing angle when the input dimension is smaller than the output dimension. Masoud et al. [18] propose an input shaping method based on phase plane analysis and the use of geometric constraints to locate the switching time of the shaper. However, this method approximates the variable rope length to a fixed rope length, which cannot analytically represent the rope length dynamics. Aiming at the operation problem of the crane in a cluttered work environment, the authors of [19] carefully designed a cubic spline curve based on the rest-to-rest strategy, which enables the crane to effectively avoid obstacles while transporting payloads. In [20], the authors apply the differential flatness theory to trajectory planning based on segmented polynomials for the obstacle avoidance transportation problem of a gantry crane with constraints.

However, none of the abovementioned methods can effectively deal with external perturbations, and especially wind disturbances. In order to solve this problem, a wind

disturbance compensation method based on a state simulator is proposed in [21], which reduces the system deviation through a proportional regulator. In [22], a method that combines the output shaping with the adaptive technique is proposed, and the adaptive algorithm is updated online to ensure the system output convergence to the output of the reference model under wind disturbance. Aiming at the finite time control problem of cranes, a method based on the flat output is proposed in [23]. The chain control law ensures the stability of the system. To improve the dynamic performance of the system, Xiao et al. [24] proposed a hybrid method that combines trajectory planning and LQR control in which the weight matrixes are searched via a multiobjective genetic algorithm. In [25], a tracking control method based on feedforward and LQR control is proposed for three-dimensional (3D) cranes. In recent years, the backstepping method has been gradually applied to the antisway control of cranes. In [26], a control method is proposed that combines fuzzy reinforcement learning with backstepping. The suppression of the payload swing is achieved by determining the parameters of the backstepping controller through reinforcement learning. In [27], a backstepping controller is designed based on the partial differential equation model, and the antisway control of the 3D crane is realized by using the boundary control scheme. In [28], a filter-based backstepping control method is proposed to ensure that the virtual control variables are only related to a strict feedback system, which ultimately ensures the convergence of the pendulum angle.

These feedback control schemes usually involve antiswing control only and do not consider the influences of the transport time or production changes in the crane system. Therefore, this paper proposes a control method that combines feedforward and feedback in which: (1) the rope length is variable to improve the transportation efficiency and safety; (2) the feedforward part builds a satisfactory trajectory that fully analyzes and summarizes the relevant constraints, and which has the advantages of high transportation efficiency and a small swing angle; (3) the backstepping method is applied to the antisway control of the underactuated crane in the feedback part with the introduction of the differential flatness theory. Without relying on complex variable substitution and control algorithms, a simple backstepping tracking controller is designed, which has a good control performance under wind disturbance.

The remainder of this paper is organized as follows: Section 2 is concerned with modeling the dynamics of a 2D overhead crane and determining a set of differential flat outputs. The analysis of and solution to the trajectory planning problem are presented in Section 3. The establishment and proof of the proposed flatness-based backstepping control scheme are given in Section 4. In Section 5, the performance of the proposed control method is compared with the traditional LQR controller to verify the effectiveness via simulations. The conclusion of the paper is given in Section 6.

## 2. Mathematical Modelling

### 2.1. Dynamic Model

The model of the two-dimensional(2D) overhead crane is shown in Figure 1. The trolley position is denoted by $x$, $l$ denotes the rope length, and $\theta$ is the payload swing angle. The payload position in the coordinate axis can be represented as $(x_p, y_p)$. $F_1$ is the force that drives the trolley forward, $F_2$ is the resultant force that drives the payload lifting, and $f_{wind}$ denotes the wind disturbance during the operation of the crane system.

**Assumption 1.** *The effect of air friction is neglected*;

**Assumption 2.** *The rope is always tense when the payload is lifted and lowered, and so the overhead crane system can be regarded as a rigid multibody system*;

**Assumption 3.** *The payload swing angle ($\theta$) satisfies $\theta(t) \in (-\pi/2, \pi/2)$*;

**Assumption 4.** *The state of the crane can be measured directly*.

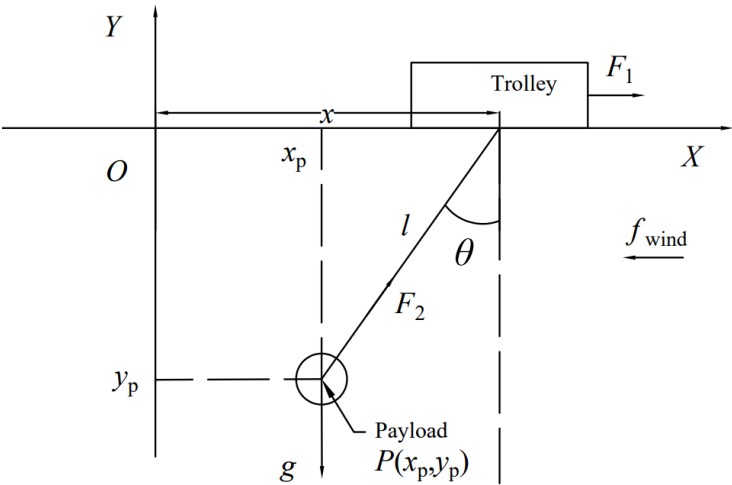

**Figure 1.** 2D overhead crane model.

Defining the generalized coordinates ($q = (x, l, \theta)^T$), the dynamic equations of the overhead crane(1)–(3) can be obtained by Lagrange equations, as follows:

$$(M + m)\ddot{x} + m\ddot{l}\sin\theta + 2m\dot{l}\dot{\theta}\cos\theta + ml\ddot{\theta}\cos\theta - ml\dot{\theta}^2\sin\theta = F_1 - f \tag{1}$$

$$m\ddot{l} + m\ddot{x}\sin\theta - ml\dot{\theta}^2 - mg\cos\theta = F_2 \tag{2}$$

$$2ml\dot{l}\dot{\theta} + mll\ddot{\theta} + ml\ddot{x}\cos\theta + mlg\sin\theta = -f_{\text{wind}} \tag{3}$$

where $M$ and $m$ denote the masses of the trolley and payload, respectively, $g$ is the gravitational constant, and $f$ denotes the friction of the rail, which can usually be expressed as:

$$f = r_1\tanh(\dot{x}/\tau) - r_2|\dot{x}|\dot{x} \tag{4}$$

where $r_1$, $r_2$, and $\tau$ are all friction-related coefficients that need to be set. Equations (1)–(3) can be expressed in the following matrix-vector form:

$$M(q)\ddot{q} + C(\dot{q}, q)\dot{q} + G(q) = U - V \tag{5}$$

where $M(q)$ is the symmetric mass matrix; $C(\dot{q}, q)$ is the Coriolis and centrifugal force matrix; $G(q) = (0, -mg\cos\theta, mgl\sin\theta)^T$ denotes the gravitational vector; $U = (F_1, F_2, 0)^T$ denotes the driving force vector; $V = (-f, 0, -f_{\text{wind}})^T$ denotes the external perturbation vector. The wind disturbance ($f_{\text{wind}}$) can be expressed by the following equation [29]:

$$\begin{aligned} f_{\text{wind}} &= cwA \\ w &= \tfrac{1}{2}\rho v^2_{\text{wind}} \end{aligned} \tag{6}$$

where $c$ is the wind coefficient, $A$ is the windward area, $w$ is the calculated wind pressure, $\rho$ is the air density, and the wind speed data ($v_{\text{wind}}$) are simulated by filtering a white noise filter using a low-pass filter ($\frac{T}{Ts+1}$) [30], where $T$ is the time constant.

### 2.2. Differential Flat Output Construction

Differential flatness refers to a nonlinear system [31] in which all the state variables and input variables can be represented algebraically by a set of flat outputs and their finite-order derivatives. Because the characteristics of the underactuated system make it difficult to control the payload swing angle, it can be transformed to control the flat output by mapping the output variable to the flat output space.

As shown in Figure 1, the payload position $(x_p, y_p)$ is defined as the flat output. It can be obtained according to the Newtonian mechanics formula and geometric analysis without considering factors such as the wind disturbance and friction force:

$$m\ddot{x}_p = R(-\sin\theta) \tag{7}$$

$$m\ddot{y}_p = R\cos\theta - mg \tag{8}$$

$$x_p = x + l\sin\theta \tag{9}$$

$$y_p = -l\cos\theta \tag{10}$$

where $R$ denotes the pulling force on the rope. Taking Equation (8) into (7) yields:

$$\theta = -\arctan\left(\frac{\ddot{x}_p}{\ddot{y}_p + g}\right) \tag{11}$$

By substituting Equations (10) and (11) into (9), we can obtain:

$$x = x_p - \frac{\ddot{x}_p y_p}{\ddot{y}_p + g} \tag{12}$$

Finally, the $l$ can be obtained from Equations (9), (10), and (12):

$$l = \sqrt{\left(\frac{\ddot{x}_p y_p}{\ddot{y}_p + g}\right)^2 + y_p^2} \tag{13}$$

## 3. Feedforward Control Design

### 3.1. Constraint Consideration

The primary control objective of the crane system is to move the trolley to the specified position while suppressing the payload swing. In order to achieve this goal, the planned trajectory must satisfy the following constraints:

Constraint 1: The trolley starts from the initial position $(x_0)$ at the moment $t = 0$, and it arrives at the desired position $(x_e)$ after a finite time $(t_e)$. The velocity and acceleration at the initial and termination moments are 0:

$$\begin{cases} x(0) = \dot{x}(0) = \ddot{x}(0) = 0 \\ x(t_e) = x_e \\ \dot{x}(t_e) = \ddot{x}(t_e) = 0 \end{cases} \tag{14}$$

Constraint 2: The maximum speed and maximum acceleration of the trolley are limited for the sake of safety during transportation:

$$\begin{cases} |\dot{x}(t)| \leq v_{max} \\ |\ddot{x}(t)| \leq a_{max} \end{cases} \tag{15}$$

Constraint 3: To ensure that the initial swing angle is 0 and that it eventually converges to 0, the boundary conditions related to the swing angle are as follows. During the operation, the maximum payload swing angle is bounded:

$$\begin{cases} \theta(0) = \dot{\theta}(0) = \ddot{\theta}(0) = 0 \\ \theta(t_e) = \dot{\theta}(t_e) = \ddot{\theta}(t_e) = 0 \end{cases} \tag{16}$$

$$|\theta(t)| \leq \theta_{max} \tag{17}$$

Constraint 4: To achieve the accurate positioning of the rope length and avoid impacts on the container trailer or straddle carrier, it is necessary to set the rope length lifting speed and acceleration to 0 at the initial and termination times. Moreover, they are bounded due to the limitation of the lifting mechanism:

$$\dot{l}(0) = \dot{l}(t_e) = \ddot{l}(t_e) = \ddot{l}(t_e) = 0 \tag{18}$$

$$\begin{cases} \left| \dot{l}(t) \right| \le v_{l,max} \\ \left| \ddot{l}(t) \right| \le a_{l,max} \end{cases} \tag{19}$$

where $v_{max}$ and $a_{max}$ are the maximum speed and maximum acceleration of the trolley, respectively, $\theta_{max}$ is the maximum swing angle of the payload, and $v_{l,max}$ and $a_{l,max}$ are the maximum lifting speed and maximum lifting acceleration of the rope length, respectively.

According to Equations (10)–(13), (15), and (17), for the constraint to hold, it must be ensured that $x_p(t)$ and $y_p(t)$ have fourth-order derivatives, which can be obtained as follows:

$$\begin{cases} x_p^{(r)}(0) = x_p^{(r)}(t_e) = 0, r = 1, 2, 3, 4 \\ x_p(0) = 0, x_p(t_e) = x_e \end{cases} \tag{20}$$

The lifting displacement trajectory of the payload is planned according to the actual demand, as shown in Figure 2. For calculation convenience, the trajectory is assumed to have symmetry, and then it can be obtained:

$$\begin{cases} y_p^{(i)}(0) = y_p^{(i)}(t_e) = 0, i = 1, 2, 3, 4 \\ y_p(0) = y_p(t_e) = -l_0, y_p(\frac{t_e}{2}) = -l_m \end{cases} \tag{21}$$

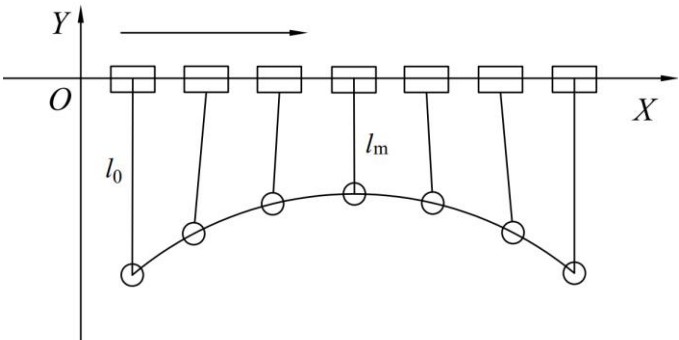

**Figure 2.** Transport process that accompanies lifting operation.

where $l_0$ and $l_m$ refer to the initial rope length and the rope length at the half-cycle moment of operation, respectively. After obtaining the set of constraints and boundary conditions associated with the differential flat output, the flat outputs of each order are next expressed in terms of parameters.

### 3.2. Trajectory Planning

Compared with other types of splines, the nonuniform B-spline curve has a better control performance because the start and end points of the curve pass through the control points. An appropriate B-spline curve is then chosen to parameterize the flat outputs and their derivatives.

Given $n + 1$ control points $(D_0, D_1, \ldots, D_n)$ and a knot vector $(S = \{s_0, s_1, \cdots, s_h\})$ that consists of $h + 1$ knots, $h = n + k + 1$. Then, a B-spline curve of the degree $(k)$ can be defined, as suggested in [32]:

$$C(s) = \sum_{i=0}^{n} D_i N_{i,k}(s), s \in [s_i, s_{i+1}] \tag{22}$$

where $N_{i,k}(s)$ is the base function of the B-spline function.

Furthermore, referring to [33], the $r$-order derivative of the B-spline curve can be defined as:

$$D_i^{(r)} = \begin{cases} D_i, r = 0 \\ \frac{k-r+1}{u_{i+k+1}-u_{i+r}}(D_{i+1}^{(r-1)} - D_i^{(r-1)}), r > 0 \end{cases} \tag{23}$$

$$C^{(r)}(s) = \sum_{i=0}^{n-r} D_i^{(r)} N_{i,k-r}(s) \tag{24}$$

It is worth noting that because $t = st_e$, the $r$-th derivative of the flat output should be expressed as follows:

$$\begin{bmatrix} x_{\mathrm{p}}^{(r)}(t) \\ y_{\mathrm{p}}^{(r)}(t) \end{bmatrix} = C^{(r)}(s)\frac{1}{t_{\mathrm{e}}^r}, r \geq 0 \tag{25}$$

B-spline curves are usually determined by control points and knot vectors. The first $k + 1$ and last $k + 1$ knots of nonuniform B-spline curves are set as equal, where $s_0 = s_1 = \cdots = s_k = 0$ and $s_{n+1} = s_{n+2} = \cdots = s_{n+k+1} = 1$, and the remaining $n - k$ internal knots can be determined by the control points according to the Hartley–Judd method [34]. The sequence of the control points of the crane system can be described as $D_i = [Dx_i, Dy_i]^{\mathrm{T}}, i = 0, 1, \ldots, n$.

By substituting Equation (25) into (20) and (21), one obtains:

$$\begin{cases} D_0 = D_1 = D_2 = D_3 = D_4 = [0, -l_0]^{\top} \\ D_n = D_{n-1} = D_{n-2} = D_{n-3} = D_{n-4} = [x_{\mathrm{e}}, -l_0]^{\top} \end{cases} \tag{26}$$

According to Equations (20) and (21), the remaining control points can be parameterized by polynomial interpolation as Equations (27) and (28):

$$Dx(i) = x_{\mathrm{e}}\{\sum_{j=0}^{9} \left[a_j \left(\frac{i-4}{n-8}\right)^j\right]\} \tag{27}$$

$$Dy(i) = \sum_{j=0}^{10} [b_j(\frac{i-4}{n-8})^j] \tag{28}$$

where $i = 5, \ldots, n-5$, $a_j$ is calculated by Equation (20) and determined by $x_{\mathrm{e}}$ and $t_{\mathrm{e}}$, and $b_j$ is calculated by Equation (20) and determined by $Dy_{\mathrm{m}}$, $l_0$, and $t_{\mathrm{e}}$. Equations (27) and (28) represent the sequence of control points rather than the payload trajectory; thus, $Dy_{\max} = Dym$ is set to change the value of the control points from Equations (20) and (21), and to ensure the continuance and smoothness of $x_{\mathrm{p}}(t)$ and $y_{\mathrm{p}}(t)$, the degree of the spline curves needs to be greater than 4. $k = 7$ and $n = 15$ were chosen, and then the differential flat output expressed by the B-spline curve could be obtained.

Generally speaking, in the process of loading and unloading the payloads of crane systems, the peak value of the swing angle is inversely proportional to the transportation time. To comprehensively consider the payload swing angle and transportation time, the objective function is designed as follows:

$$F(D_{ym}, t_{\mathrm{e}}) = v_1|\theta|_{\max} + v_2 t_{\mathrm{e}} \tag{29}$$

where $v_1$ and $v_2$ are the weights to be given. The optimization problem is solved by PSO: min $F(Dy_m, t_e)$, s.t. Equations (15), (17), and (19). The desired flat output $(x_{pd}, y_{pd})$ is obtained by substituting the optimal parameters into Equation (25). The optimal trajectories $(x_d, l_d,$ and $\theta_d)$ can finally be obtained by substituting $(x_{pd}, y_{pd})$ into Equations (11)–(13).

**Remark 1.** *In this algorithm design, $Dy_m$ and $t_e$ represent particle populations that are constantly changing positions. By continuously updating the velocities and positions of the particles in the population, the global optimal solution of the entire particle swarm can finally be obtained, where $Dy_m$ denotes the minimum value of the $Dy$ in the control point sequence (D). Because the curve falls in the convex hull of the control point sequence, the value of the $Dy_m$ is slightly smaller than the lm.*

## 4. Trajectory Tracking Control

Although the obtained optimal trajectory has an excellent performance, including a short transportation time and small payload swing angle, the control becomes less effective under external disturbances, such as wind disturbances. Therefore, the control effect of crane systems can be improved using tracking controllers. The ideal tracking controller should make $x(t) \to x_d(t)$, $l(t) \to l_d(t)$, and $\theta(t) \to \theta_d(t)$, but according to Equation (5), the motion of the trolley, lifting of the rope, and payload swing are highly coupled, which is challenging to the control of the system state. To solve this problem, a backstepping controller based on differential flatness was designed. The overall control flow is shown in Figure 3.

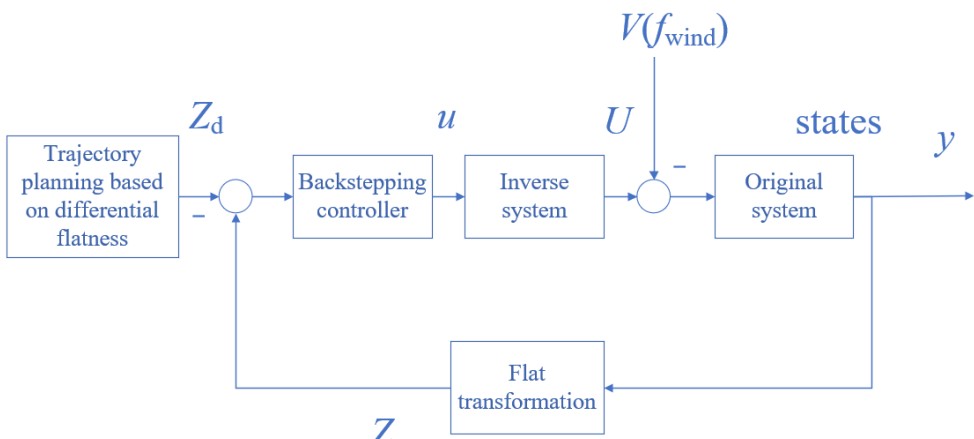

**Figure 3.** Control flow chart.

As shown in Figure 3, the entire control process is to achieve feedback control under external disturbances through the backstepping method after obtaining the ideal differential flat output. The actual control force can be obtained after the output of the backstepping controller is transformed by the inverse system. The specific control objectives are shown in Equation (30):

$$\lim_{t\to\infty}[z_1\ z_2\ z_3\ z_4] = [z_{d1}\ z_{d2}\ z_{d3}\ z_{d4}] \tag{30}$$

where $z_1, z_2, z_3,$ and $z_4$ are shown in Equation (31), while $z_{d1}, z_{d2}, z_{d3},$ and $z_{d4}$ are shown in Equation (32):

$$
\begin{aligned}
z_1 &= [x_{pc}, y_{pc}]^\top, z_2 = [\dot{x}_{pc}, \dot{y}_{pc}]^\top \\
z_3 &= [\ddot{x}_{pc}, \ddot{y}_{pc}]^\top, z_4 = [x_{pc}^{(3)}, y_{pc}^{(3)}]^\top
\end{aligned}
\tag{31}
$$

$$
\begin{aligned}
z_{d1} &= [x_{pd}, y_{pd}]^\top, z_{d2} = [\dot{x}_{pd}, \dot{y}_{pd}]^\top \\
z_{d3} &= [\ddot{x}_{pd}, \ddot{y}_{pd}]^\top, z_{d4} = [x_{pd}^{(3)}, y_{pd}^{(3)}]^\top
\end{aligned}
\tag{32}
$$

where $x_{\text{pc}}^{(i)}$ and $y_{\text{pc}}^{(i)}$, $i = 0, \ldots, 4$, are the actual differential flat outputs obtained by the backstepping and inverse transformation of the original system. $x_{\text{pd}}^{(i)}$ and $y_{\text{pd}}^{(i)}$, $i = 0, \ldots, 4$, are the ideal flat outputs designed by the feedforward section.

A chained system is defined according to the flat output, as follows:

$$\begin{aligned}
\dot{z}_1 &= z_2 \\
\dot{z}_2 &= z_3 \\
\dot{z}_3 &= z_4 \\
\dot{z}_4 &= u
\end{aligned} \tag{33}$$

where $u$, which is shown in Equation (34), denotes the new control input for this system, which can be obtained when the control law is designed according to the backstepping method.

$$u = [x_{\text{b}}^{(4)}, y_{\text{b}}^{(4)}] \tag{34}$$

To deal with the explosion of the complexity problem of the backstepping methods, a set of first-order filters is constructed, as follows [35]:

$$\beta_i \dot{z}_{i+1,v} + z_{i+1,v} = \alpha_i \tag{35}$$

where $i = 1,2,3$, $\alpha_i$ is the virtual control law to be designed, $\beta_i \in \mathbb{R}^2$ is a small positive constant vector, and $z_{i+1,v}(0) = \alpha_i(0)$. $\gamma_i$, $i = 1,2,3$, denotes the filtering error, as follows:

$$\gamma_i = z_{i+1,v} - \alpha_i \tag{36}$$

**Step 1.** Define the tracking error of the flat output, as shown in Equation (37):

$$e_1 = z_1 - z_{d1} \tag{37}$$

The Lyapunov function is chosen, as follows:

$$V_1 = \frac{1}{2} e_1^\top e_1 \tag{38}$$

Taking the derivative of both sides, we can obtain:

$$\dot{V}_1 = e_1^\top \dot{e}_1 = e_1^\top (z_2 - z_{\text{d2}}) \tag{39}$$

Construct the virtual variable ($\alpha_1$), as shown in Equation (40), as the control input, and:

$$\alpha_1 = z_{\text{d2}} - k_1 e_1 \tag{40}$$

where $k_1 \in \mathbb{R}^{2 \times 2}$ is a positive definite diagonal constant matrix. The error ($e_2$) between the virtual control variable ($z_{2,v}$) and $z_2$ is shown as follows:

$$e_2 = z_2 - z_{2,v} \tag{41}$$

By substituting Equations (36), (40), and (41) into (39), we obtain:

$$\dot{V}_1 = -e_1^\top k_1 e_1 + e_1^\top (e_2 + \gamma_1) \tag{42}$$

**Step 2.** To make the $\dot{V}_1$ negative semidefinite, construct a new Lyapunov function, as follows:

$$V_2 = V_1 + \frac{1}{2} e_2^\top e_2 \tag{43}$$

The derivative of Equation (43) with respect to time is given by:

$$\dot{V}_2 = -e_1^\top k_1 e_1 + e_1^\top (e_2 + \gamma_1) + e_2^\top \dot{e}_2 \tag{44}$$

Take the derivative of both sides of Equation (41) and substitute Equations (35) and (36) into it:

$$\dot{e}_2 = \dot{z}_2 - \dot{z}_{2,v} = z_3 - \frac{\alpha_1 - z_{2,v}}{\beta_1} = z_3 + \frac{\gamma_1}{\beta_1} \tag{45}$$

The error ($e_3$) between the virtual control variable ($z_{3,v}$) and $z_2$ is as follows:

$$e_3 = z_3 - z_{3,v} \tag{46}$$

Substitute Equations (36), (45) and (46) into (44):

$$\dot{V}_2 = -e_1^\top k_1 e_1 + e_1^\top (e_2 + \gamma_1) + e_2^\top (e_3 + \alpha_2 + \gamma_2 + \frac{\gamma_1}{\beta_1}) \tag{47}$$

The virtual controller ($\alpha_2$) is constructed as follows:

$$\alpha_2 = -\frac{\gamma_1}{\beta_1} - k_2 e_2 - e_1 \tag{48}$$

where $k_2 \in \mathbb{R}^{2\times 2}$ is the positive definite diagonal constant matrix.

By substituting Equations (48) into (47), one has:

$$\dot{V}_2 = -e_1^\top k_1 e_1 - e_2^\top k_2 e_2 + e_1^\top \gamma_1 + e_2^\top (e_3 + \gamma_2) \tag{49}$$

**Step 3.** Similarly, in order to make the $\dot{V}_2$ negative semidefinite, construct a new Lyapunov function ($V_3$), as follows:

$$V_3 = V_2 + \frac{1}{2} e_3^\top e_3 \tag{50}$$

The time derivative of $V_3$ is given by:

$$\dot{V}_3 = -e_1^\top k_1 e_1 - e_2^\top k_2 e_2 + e_1^\top \gamma_1 + e_2^\top (e_3 + \gamma_2) + e_3^\top \dot{e}_3 \tag{51}$$

Take the derivative of both sides of Equation (46) and substitute Equations (35) and (36) into it:

$$\dot{e}_3 = \dot{z}_3 - \dot{z}_{3,v} = z_4 + \frac{\gamma_2}{\beta_2} \tag{52}$$

It is expected that $z_4$ can track $z_{4,v}$; thus, the tracking error ($e_4$) is constructed as:

$$e_4 = z_4 - z_{4,v} \tag{53}$$

Substitute Equations (36), (52), and (53) into (51):

$$\dot{V}_3 = -e_1^\top k_1 e_1 - e_2^\top k_2 e_2 + e_1^\top \gamma_1 + e_2^\top (e_3 + \gamma_2) + e_3^\top (e_4 + \alpha_3 + \gamma_3 + \frac{\gamma_2}{\beta_2}) \tag{54}$$

Construct the virtual variable ($\alpha_3$) as a new control input:

$$\alpha_3 = -\frac{\gamma_2}{\beta_2} - k_3 e_3 - e_2 \tag{55}$$

where $k_3 \in \mathbb{R}^{2\times 2}$ is a positive definite diagonal constant matrix. By substituting Equation (55) into (54), the $\dot{V}_3$ can be obtained, as follows:

$$\dot{V}_3 = -e_1^\top k_1 e_1 - e_2^\top k_2 e_2 - e_3^\top k_3 e_3 + e_1^\top \gamma_1 + e_2^\top \gamma_2 + e_3^\top (e_4 + \gamma_3) \tag{56}$$

**Step 4.** Similarly, let the Lyapunov function ($V_4$) be constructed as follows so that $\dot{V}_3$ can be negative semidefinite:

$$V_4 = V_3 + \frac{1}{2} e_4^\top e_4 \tag{57}$$

The time derivative of $V_4$ can be written as:

$$\dot{V}_4 = -e_1^\top k_1 e_1 - e_2^\top k_2 e_2 - e_3^\top k_3 e_3 + e_1^\top \gamma_1 + e_2^\top \gamma_2 + e_3^\top (e_4 + \gamma_3) + e_4^\top \dot{e}_4 \tag{58}$$

Take the derivative of both sides of Equation (53) and substitute Equations (35) and (36) into it:

$$\dot{e}_4 = \dot{z}_4 - \dot{z}_{4,v} = u + \frac{\gamma_3}{\beta_3} \tag{59}$$

Considering Equations (58) and (59), to make $\dot{V}_4$ negative semidefinite, the control input ($u$) is designed as:

$$u = -\frac{\gamma_3}{\beta_3} - k_4 e_4 - e_3 \tag{60}$$

where $k_4 \in \mathbb{R}^{2\times2}$ is a positive definite diagonal constant matrix. By substituting Equations (59) and (60) into (58), one has:

$$\dot{V}_4 = -e_1^\top k_1 e_1 - e_2^\top k_2 e_2 - e_3^\top k_3 e_3 - e_4^\top k_4 e_4 + e_1^\top \gamma_1 + e_2^\top \gamma_2 + e_3^\top \gamma_3 \tag{61}$$

By selecting the suitable $\beta_i$, the $V_4$ is positive definite, and the $\dot{V}_4$ can be negative definite at the nonzero solution. According to the Lyapunov stability principle, the $e_4$ is asymptotically stable. Using forward recursion, it can be found that the $e_1$, $e_2$, and $e_3$ are also asymptotically stable. Then, the whole system can be stabilized.

In Equation (34), the control input is the fourth-order derivative, while the flat initial value of each order is 0. The flat derivative of each order designed by the backstepping method can be obtained by integrating Equation (60):

$$\begin{aligned} z_{b1} = [x_b, y_b]^\top, z_{b2} = [\dot{x}_b, \dot{y}_b]^\top \\ z_{b3} = [\ddot{x}_b, \ddot{y}_b]^\top, z_{b4} = [x_b^{(3)}, y_b^{(3)}]^\top \end{aligned} \tag{62}$$

From Equations (11)–(13), each state of the crane system can be represented by their derivatives. Then, taking $f$ and $f_{\text{wind}}$ as the disturbances, according to Equations (1) and (3), the control forces ($F_1$ and $F_2$) can be similarly represented by the differential flat outputs ($z_{b1}$, $z_{b1}$, $z_{b2}$, $z_{b3}$, $z_{b4}$, and $u$). In this way, the inverse dynamic system can be set up as follows:

$$F_1 = \phi(z_{b1}, z_{b2}, z_{b3}, z_{b4}, u) \tag{63}$$

$$F_2 = \varphi(z_{b1}, z_{b2}, z_{b3}, z_{b4}, u) \tag{64}$$

By substituting Equations (63) and (64) into (5) as the inputs, the states of the crane ($[x, l, \theta]^\top$) and their derivatives can be obtained. The actual differential flat output of each order ($z_1$, $z_2$, $z_3$, and $z_4$) can be obtained by the definition of the flat outputs.

**Remark 2.** *Due to the underactuated characteristics of the crane system, there is no force that can directly control the payload swing angle, which makes the backstepping controller difficult to design. In order to solve this problem, the differential flat theory is introduced, and the indirect control of the system state is realized by the control of the flat output. While the actual differential flat output is shown in Assumption 4, when the state of the crane system can be directly measured, it is obtained from the system state through the inverse transformation of Equations (9) and (10).*

## 5. Simulation Results

The effectiveness of the proposed method was verified by running simulations under Matlab/Simulink. The control performance was confirmed by observing the trolley displacement, rope length variation, and payload swing angle curves. First, the trajectory planning method in the third section was verified. Considering no wind disturbance and friction, the crane parameters and optimization constraints are summarized in Table 1. After the algorithm solution, the optimal parameters were $Dy_m$ = 0.41 m, $t_e$ = 7.8589 s.

**Table 1.** Parameters of 2D overhead crane.

| Parameters | Values |
|---|---|
| $M$ | 10 (kg) |
| $m$ | 1 (kg) |
| $g$ | 9.8 (m/s$^2$) |
| $l_0$ | 0.7 (m) |
| $x_e$ | 0.7 (m) |
| $v_{l,max}$ | 1.5 (m/s) |
| $a_{l,max}$ | 0.75 (m/s$^{2)}$) |
| $v_{max}$ | 0.2 (m/s) |
| $a_{max}$ | 0.5 (m/s$^2$) |
| $v_1$ | 9.5 |
| $v_2$ | 0.17 |

From Figure 4, it can be seen that the trajectory of the system meets the constraint conditions and design requirements. At this time, the maximum payload swing angle is 0.55°, which means that the method has a good effect on suppressing the payload swing.

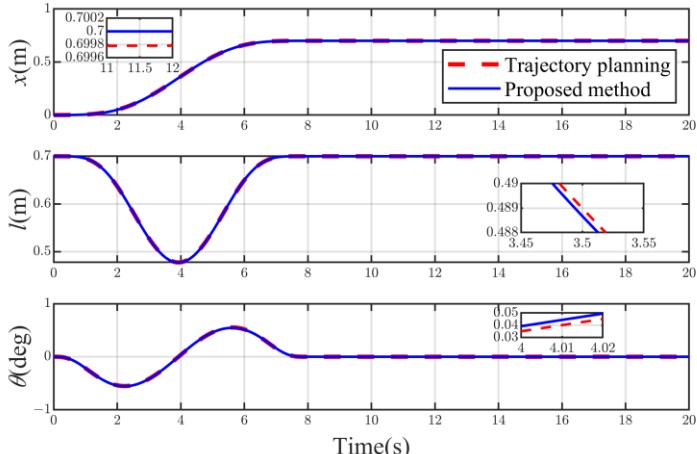

**Figure 4.** Trolly displacement, rope length, and payload swing angle without external disturbances.

The backstepping gains based on the flat output can be seen in Table 2. The solid blue lines in Figures 4 and 5 show the tracking capability of the proposed method without external perturbations. The designed tracking controller can accurately and rapidly track the desired trajectory. During the operation of the trolley, the payload rises and falls according to the desired trajectory, and the swing angle finally converges to zero.

**Table 2.** Parameters of proposed backstepping controller.

| Parameters | Values |
|---|---|
| $k_1$ | *diag* $\{8, 5\}$ |
| $k_2$ | *diag* $\{10, 5\}$ |
| $k_3$ | *diag* $\{8, 5\}$ |
| $k_4$ | *diag* $\{8, 5\}$ |
| $\beta_1$ | $[0.01, 0.01]^T$ |
| $\beta_2$ | $[0.01, 0.01]^T$ |
| $\beta_3$ | $[0.01, 0.01]^T$ |

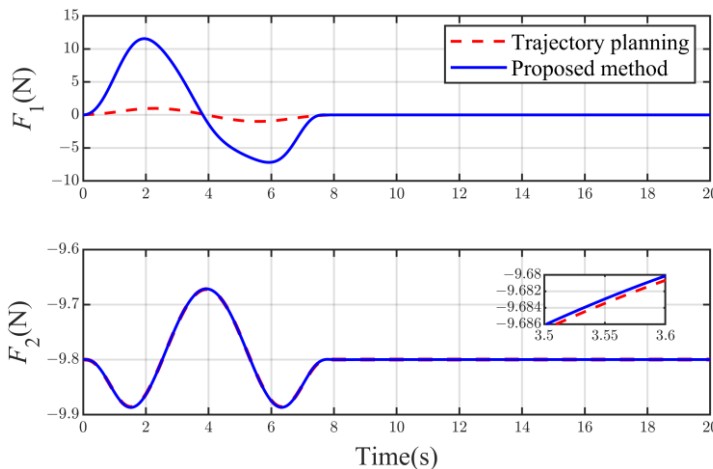

**Figure 5.** Control input without external disturbances.

Then, in order to verify the performance of the backstepping tracking controller, the initial positions of the trolley and rope length were changed to −0.01 m and 0.72 m, respectively. As shown in Figure 6, the proposed controller responds rapidly and forces the system state to accurately track the reference trajectory after 2.64 s, while the trajectory planning method does not have this ability, which is also verified by the control input curve in Figure 7.

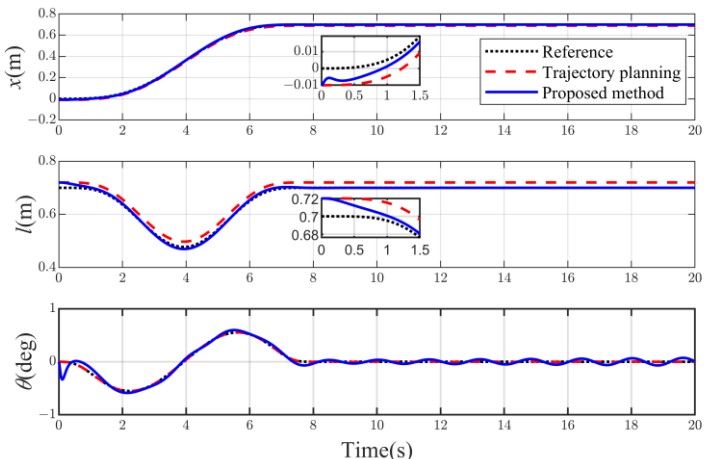

**Figure 6.** System state with initial deviation.

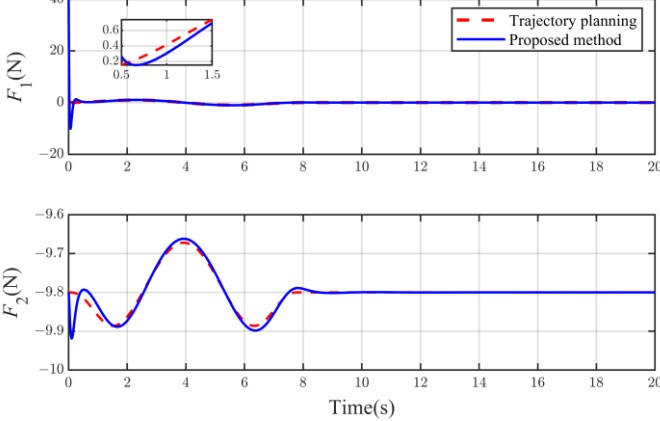

**Figure 7.** Control input with initial deviation.

In the next step, friction force (4) and wind disturbance (6) are introduced as the external disturbances, and the time constant ($T$) is set as 20 s. The simulated wind disturbance is shown in Figure 8. The wind disturbance fluctuated between 0.032 N and 0.038 N, which is consistent with the wind disturbance waveform in the literature [30]. The wind blew perpendicular to the 90° direction to the payload in the simulation so that the wind disturbance was always at its maximum.

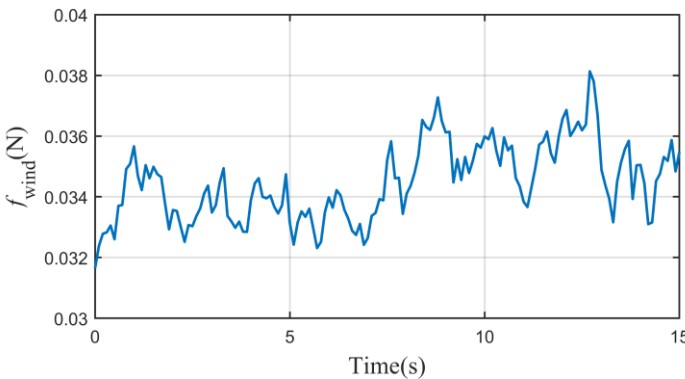

**Figure 8.** Simulated wind disturbance.

After obtaining the wind disturbance data, in order to demonstrate the anti-interference ability of the proposed method, it was compared with the LQR method in [11]. The LQR controller is designed for system (33), as follows:

$$u = z_{d5} - \eta_1(z_1 - z_{d1}) - \eta_2(z_2 - z_{d2}) - \eta_3(z_3 - z_{d3}) - \eta_4(z_4 - z_{d4}) \tag{65}$$

where $z_{d5} = \left[x_p^{(4)}, y_p^{(4)}\right]^\top$ is the differential flat fourth derivative obtained from the trajectory planning, and the resulting states and inputs are substituted into Equations (63) and (64) to obtain the actual control force ($[F_1, F_2]^T$). The control gains of the LQR can be seen in Table 3.

**Table 3.** Parameters of LQR controller.

| Parameters | Values |
|:---:|:---:|
| $\eta_1$ | $[12.247, 10]^T$ |
| $\eta_2$ | $[32.675, 25.086]^T$ |
| $\eta_3$ | $[37.463, 26.466]^T$ |
| $\eta_4$ | $[18.026, 12.367]^T$ |

Figures 9 and 10 show the antisway control effects of the different methods under external disturbances. As shown in Figure 6, all the methods performed the payload lift and low operations well, namely $l \rightarrow l_d$. Both the LQR and the proposed method completed the positioning operation of the trolley, namely $x \rightarrow x_d = 0.7\text{m}$. In addition, due to the influence of the wind disturbance, all the methods had payload swing deflection angles of $-0.289°$. Trajectory planning is an open-loop method, and there is no way to deal with disturbances; thus, the trolley could not be stabilized at the specified position ($x_d = 0.7$ m), and the payload swung violently at this time, with a maximum swing angle of $1.1474°$ and a residual oscillation of $0.3959°$. The maximum swing angle of the LQR method is $0.8962°$, and the residual oscillation is less than $0.0243°$. In contrast, the maximum swing angle of the proposed method is $0.8715°$, and the residual oscillation is less than $0.0122°$. Therefore, the proposed method has the same control performance as the LQR method in terms of trolley positioning, but it is better than the trajectory planning and LQR methods in terms of the antisway control under wind disturbance.

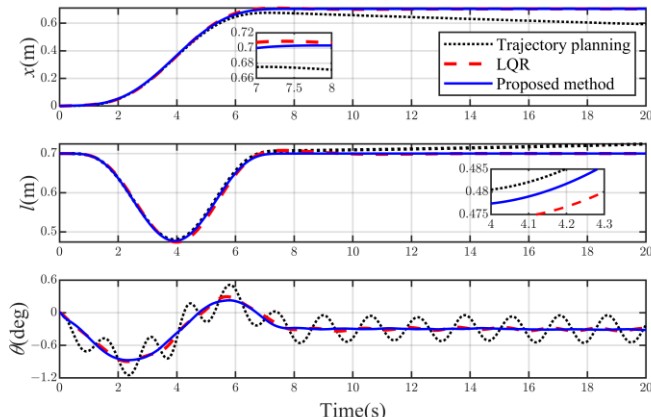

**Figure 9.** System state under external disturbances.

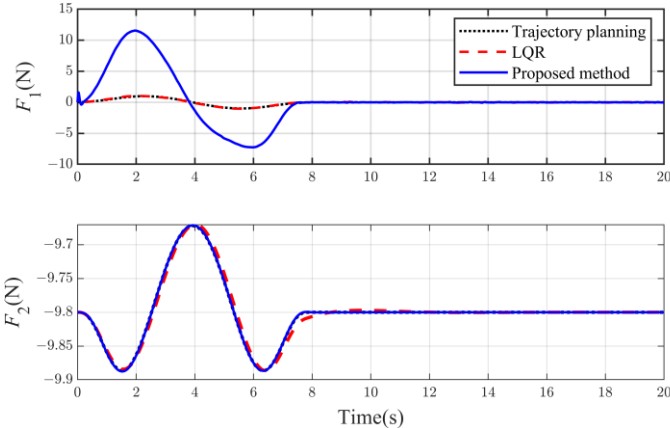

**Figure 10.** Control input under external disturbances.

## 6. Conclusions

In this paper, the differential flatness theory was applied to the design of feedforward and feedback control to solve the antisway problem of a 2D underactuated overhead crane. First, to obtain a satisfactory trajectory, the B-spline function is used to parameterize the flat output, and the optimal parameters that conform to the constraint are obtained through an algorithm. Next, a backstepping controller is designed by defining a strict feedback system based on a flat output space. The differential flatness feature ensures the free transformation between the flat outputs and crane states. Compared with the trajectory planning and LQR methods, the proposed control scheme can accurately position the trolley and suppress the payload swing when the rope length changes, and it has strong robustness under wind disturbance. The following work will consist of the experimental validation and will consider the application of the observer to the proposed method.

**Author Contributions:** All the authors discussed the idea, conducted the theoretical research, and formulated the problem. Conceptualization, W.N.; methodology, Z.Y.; simulation, Z.Y.; writing—original draft, Z.Y.; writing—review and editing, W.N. and Z.Y.; supervision, W.N. and Z.Y. All authors have read and agreed to the published version of the manuscript.

**Funding:** This research was funded by the Capacity Building Program of the Municipal Universities of Shanghai (20040501400).

**Data Availability Statement:** No new data were created.

**Conflicts of Interest:** The authors declare no conflict of interest.

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
