# Peer review of "Flatness-Based Backstepping Antisway Control of Underactuated Crane Systems under Wind Disturbance"

_electronics, doi:10.3390/electronics12010244_

Round 1
Reviewer 1 Report
1. Line 100, page 2, “f is the friction coefficient…” should be “f is the friction…”.
2. The system (33) is a linear system instead of a strict feedback system mentioned on line 215, page 7.
3. The control law in (56) contains $e_1^{(3)},e_2^{(2)},e_3^{(1)}” which may cause difficulty in implementation, i.e., the so-called explosion of complexity problem. The authors need to clarify this issue.
4. Since the system (33) is a linear system, it is not clear why not use the standard linear state feedback control.
Reviewer 2 Report
The proposed work is devoted to the analysis of compensation of crane vibrations in air flows. Both the theoretical part and the experiment of the simulation model are considered. In addition, a preliminary review was conducted, where the relevance of this study was determined. In comparison with other works, a more detailed analysis of all the interference that can be compensated is made here. The mathematical model has shown its adequacy in experimental analysis. The conclusion is adequate, all conclusions answer the tasks set. The references, in turn, cover the works of other authors in this field of research. There are several points in the work, this is an incorrect display of characters in line 222, 229, 231, 239, 247. In general, I consider the work ready for publication.
Reviewer 3 Report
1. Fig. 4 - red dashed line same as a blue line. maybe use a thicker red line? or non-linear log scale for Y? please show the difference between lines
2. Fig. 10 same
3. equations 27, 28 - range of j is different, why? please check equations or few words about why it so.
Round 2
Reviewer 1 Report
The previous comments have been well addressed. I have no further comments.